# Offline Policy Optimization with Eligible Actions

**Yao Liu**[*1]        **Yannis Flet-Berliac**[2]        **Emma Brunskill**[2]

[1]ByteDance, yao.liu.chn@gmail.com
[2]Stanford University, {yfletberliac, ebrun}@cs.stanford.edu

## Abstract

Offline policy optimization could have a large impact on many real-world decision-making problems, as online learning may be infeasible in many applications. Importance sampling and its variants are a common used type of estimator in offline policy evaluation, and such estimators typically do not require assumptions on the properties and representational capabilities of value function or decision process model function classes. In this paper, we identify an important overfitting phenomenon in optimizing the importance weighted return, in which it may be possible for the learned policy to essentially avoid making aligned decisions for part of the initial state space. We propose an algorithm to avoid this overfitting through a new per-state-neighborhood normalization constraint, and provide a theoretical justification of the proposed algorithm. We also show the limitations of previous attempts to this approach. We test our algorithm in a healthcare-inspired simulator, a logged dataset collected from real hospitals and continuous control tasks. These experiments show the proposed method yields less overfitting and better test performance compared to state-of-the-art batch reinforcement learning algorithms.

## 1 INTRODUCTION

There has been a recent surge in interest, from both the theoretical and algorithmic perspective, in offline/batch Reinforcement Learning (RL). This area could potentially bring insights from RL to the growing number of application settings which produce such datasets (like healthcare [Gottesman et al., 2019, Nie et al., 2021], customer marketing [Bottou et al., 2013, Thomas et al., 2017] or home

automation [Emmons et al., 2022]), provide ways to leverage vast amounts of observational training data encoded in videos [Chang et al., 2020, Schmeckpeper et al., 2021], or advance our core understanding of the data characteristics needed to learn near optimal policies.

Many of the settings that might benefit most from offline RL, like healthcare, education or autonomous driving, may not be Markov in the available observed features, and also may not include explicit known representations of the behavior decision policy. This has inspired work on offline policy evaluation estimation methods that make minimal assumptions on the data generation process, such as importance sampling (IS) and doubly robust estimation methods [Precup et al., 2000, Jiang and Li, 2016a, Thomas and Brunskill, 2016] and offline policy learning methods that leverage such estimators [Huang and Jiang, 2020, Cheng et al., 2020, Thomas et al., 2019].

Unfortunately, we show that offline policy selection or learning algorithms that rely on such offline estimation methods that leverage IS can suffer from a key flaw. In brief, the structure of the policy estimator is such that high estimated performance can be achieved both by policies that have high average performance, and by policies that systematically avoid taking actions taken in the dataset for initial states that lead to low rewards. This can substantially inflate the estimated performance of a potential decision policy. As an intuition, consider a setting where some patients arrive much sicker than some healthier patients. In this setting, any policy for the sicker patients will likely yield slightly worse outcomes than the average outcomes for healthier patients, but a policy's value must be taken in expectation over all patients, not just the healthy patients. We detail how a number of methods, including those that have been proposed before for other reasons, do not solve this issue, including: using a validation set, shifting the reward baseline, and leveraging thresholds on an estimated behavior policy.

Fortunately, we show that a relatively simple method for constraining the policy class considered with off policy

---

[*]This work is mostly done when the author was at Stanford.

*Accepted for the 38th Conference on Uncertainty in Artificial Intelligence* (UAI 2022).

learning can greatly ameliorate the problem of propensity overfitting. Our approach can be viewed as related to pessimistic offline RL in Markov decision processes, which has relations to the robust MDP literature [Nilim and El Ghaoui, 2005] and has been receiving growing attention (e.g., Liu et al. [2020], Yu et al. [2020], Kidambi et al. [2020]). One of the key tenants of pessimistic offline RL is to maintain precise quantification over the uncertainty over the model parameters and/or value functions of the Markov decision process, given the available data. A key challenge is how to quantify statistical uncertainty when the state space is extremely large or continuous. This issue is perhaps even more paramount in offline RL settings when we wish to leverage IS-based estimators in order to make minimal assumptions over the data generation process. We show here that constraining the policy class per state only to actions taken in the data for nearby states, which may be viewed as a loose analogy to count-based uncertainty, is sufficient to lower bound the amount of propensity overfitting that can occur. Our approach still ensures asymptotic consistency of the estimation of any policy covered by the behavior policy[1] while providing significant benefits in the finite regime, by essentially constraining policies to observed actions. In this way, our method is related to other methods that revert back to the behavior policy given minimal data for MDP model or value function learning [Satija et al., 2021] or in the bandit setting [Sachdeva et al., 2020, Brandfonbrener et al., 2020]. To our knowledge, our work is the first to explore this for large, non-Markovian stochastic decision processes using policy search methods. We show in simulations and in a real dataset on patient sepsis outcomes that our approach learns policies with significantly higher expected rewards than prior methods, and that those estimates are expected to be reliable, with solid effective samples sizes, a measure of how much of the behavior data would match the proposed policy. Our results highlight and remedy a potential reliability barrier for offline RL with minimal data process and realizability assumptions.

## 2 OFFLINE POLICY OPTIMIZATION

We study the problem of offline policy optimization in sequential decision-making under uncertainty. Let the environment be a finite-horizon Contextual Decision Process (CDP) [Jiang et al., 2017]. A CDP can capture more general, non-Markovian settings (also sometimes referred to as a Non-Markov DP [Kallus and Uehara, 2019b]). A CDP is defined as a tuple $\langle \mathcal{X}, \mathcal{A}, H, P, R \rangle$, where $\mathcal{X}$ is the context space, $\mathcal{A}$ is the action space, and $H$ is the horizon. $P = \{P_h\}_{h=1}^{H}$ is the unknown transition model, where $P_h : (\mathcal{X} \times \mathcal{A})^{h-1} \rightarrow \Delta(\mathcal{X})$ is the distribution over next context given the history. $P_1 : \Delta(\mathcal{X})$ is the initial context

distribution. Similarly, $R = \{R_h\}_{h=1}^{H}$ is the reward model and $R_h : (\mathcal{X} \times \mathcal{A})^h \rightarrow \Delta([-R_{\max}, R_{\max}])$.

In this paper, we focus on learning policies that map from the most recent context to an action distribution, $\pi : \mathcal{X} \rightarrow \Delta(\mathcal{A})$. This is optimal when the domain is Markov and can often be more interpretable and more feasible to optimize given finite data in the offline setting. In offline policy optimization settings, we have a dataset with n trajectories collected by a fixed *behavior* policy $\mu : \mathcal{X} \rightarrow \Delta(\mathcal{A})$, and we aim to find a policy $\pi$ in a policy class $\Pi$ with the highest value.

Policy gradient and optimization approaches do not rely on a Markov assumption on the underlying domain, and have had some encouraging success in offline RL [Chen et al., 2019]. Often these methods leverage an importance sampling (IS) estimator in policy evaluation: $\hat{v}_{\text{IS}}(\pi) = \frac{1}{n} \sum_{i=1}^{n} \left( \sum_{h=1}^{H} r_h^{(i)} \right) \prod_{h=1}^{H} \left( \frac{\pi(a_h^{(i)}|x_h^{(i)})}{\mu(a_h^{(i)}|x_h^{(i)})} \right)$. The IS estimator is an unbiased and consistent estimate of the value under the following two assumptions:

**Assumption 1** (Overlap). *For any $\pi \in \Pi$, and any $x \in \mathcal{X}$, $a \in \mathcal{A}$, $\frac{\pi(a|s)}{\mu(a|s)} < \infty$.*

**Assumption 2** (No Confounding / Sequential ignorability). *For any policy $\pi \in \Pi$ and $\mu$, conditioning on the current context $x_h$, the sampled action $a_h$ is independent of the outcome $r_{h:H}$ and $x_{h+1:H}$.*

IS often suffers from high variance, which has prompted work into extensions such as doubly robust methods [Jiang and Li, 2016b, Thomas and Brunskill, 2016] and/or methods that balance variance and bias. Truncating the weights and using self-normalization has been shown to be empirically beneficial both in bandit and RL settings [Swaminathan and Joachims, 2015b, Futoma et al., 2020]: we refer to this as Self-Normalized Truncated IS (SNTIS):

$$\hat{v}_{\text{SNTIS}}(\pi) := \frac{\sum_{i=1}^{n} \left( \sum_{h=1}^{H} r_h^{(i)} \right) \min\left\{ \prod_{h=1}^{H} W_h^{(i)}, M \right\}}{\sum_{i=1}^{n} \min\left\{ \prod_{h=1}^{H} W_h^{(i)}, M \right\}}, \tag{1}$$

where

$$W_h^{(i)} := \frac{\pi(a_h^{(i)}|x_h^{(i)})}{\mu(a_h^{(i)}|x_h^{(i)})} \tag{2}$$

and $M$ is a constant that truncates the weights. For ease of notation in the rest of this paper, we define: $W_{1:h}^{(i)} := \prod_{h=1}^{h} W_h^{(i)}$, $W^{(i)} := W_{1:H}^{(i)}$, $W = \sum_{i=1}^{n} W^{(i)}$, and $r^{(i)} = \sum_{h=1}^{H} r_h^{(i)}$.

While this estimate can be used as a direct objective for off policy learning, it may still have a significant variance which is important when comparing across policies. Prior work in contextual bandits [Swaminathan and Joachims, 2015a,b]

---

[1]Coverage is a minimal requirement for all IS methods to be consistent estimators of a new proposed policy.

included a variance penalization in the objective based on the empirical Bernstein's inequality.

Here we provide a simple extension to the multi-step setting to yield a target objective for offline policy learning:

$$\arg\max_{\pi \in \Pi} \hat{v}_{\text{SNTIS}}(\pi) - \lambda \sqrt{\widehat{\text{Var}}(\hat{v}_{\text{SNTIS}}(\pi))}. \quad (3)$$

# 3 RELATED WORK

There is increasing interest in multi-armed bandits and offline RL to avoid overly optimistic estimates of policies computed from finite datasets that can cause suboptimal policy learning. In this paper, we will show a particular unaddressed issue with IS methods avoiding initial states that lead to poor outcomes. In contrast, prior work has shown how to use self-normalized IS (also known as weighted IS) to address over maximizing bandit rewards [Swaminathan and Joachims, 2015b]. Counterfactual risk minimization [Swaminathan and Joachims, 2015a, Joachims et al., 2018] uses variance regularization based on the empirical Bernstein's inequality for bandit problems. However, this penalization is at the policy level. Both self-normalized IS (SNIS) and variance penalization do not directly solve the problem with avoiding contexts with low reward. In Figure 2c in Appendix A, we show the counterfactual risk minimization regularization with or without self-normalization requires a large dataset to perform well. Recent work [Brandfonbrener et al., 2020] discussed a similar overfitting issue as we describe and compared the performance of offline policy optimization and model/value-based method on such issue in the bandit setting. Those authors primarily focus on the negative result of the policy optimization approach and the advantage of the model/value-based method, whereas our approach suggests a method for addressing this issue in policy optimization and focuses on the RL setting. Doubly robust estimators [Dudík et al., 2011, Jiang and Li, 2016b, Thomas and Brunskill, 2016, Kallus and Uehara, 2019a,b] have multiple benefits but, as long as the learned $Q$-function is imperfect, the issue of avoiding low performing contexts can still remain as the methods may overfit to the high/positive residual $r - Q$. Pessimism under uncertainty approaches are promising [Kidambi et al., 2020, Yu et al., 2020] but have so far only been developed for Markov settings and are not robust to model class misspecification, unlike IS-based policy optimization.

Another line of offline batch policy optimization constrains the policy search space to be close to the behavior policy, or requires the action taken to have some minimum probability under the behavior policy [Kumar et al., 2019, Buckman et al., 2020, Sachdeva et al., 2020, Fujimoto et al., 2019, Futoma et al., 2020, Liu et al., 2019, 2020]. This work has focused on algorithms and analysis for the Markov setting with additional model realizability and/or closure assumptions that are hard to verify. As we will discuss and

empirically validate later, such constraints on the expected or observed empirical behavior policy are not yet sufficient, at least in large state spaces.

Our work can be viewed as following in the recent line of work on pessimism under uncertainty Liu et al. [2020], Yu et al. [2020], Kidambi et al. [2020], Buckman et al. [2020], but adapted to provide policy search based offline learning method that does not require the Markov assumption or model realizability, and achieves strong performance given a finite dataset.

# 4 STATE PROPENSITY OVERFITTING

We identify an important potential issue with using IS estimators during offline policy optimization, as we illustrate in contextual bandits when using the SNIS estimator.

Let $v^\pi(x) = \mathbb{E}[r|x, a \sim \pi]$, $\hat{p}(x)$ be the empirical probability mass/density over the contexts $x$ in the dataset, and $W(x) = \sum_{i:x^{(i)}=x} \frac{W^{(i)}}{W}$ where $W$ is the importance weight (Eq. 2). We now decompose the importance weighted off-policy estimator into three parts.

$$\hat{v} = \underbrace{\mathbb{E}_{\hat{p}}[v^\pi(x)]}_{\text{empirical } v} + \underbrace{\sum_{x \in \mathcal{X}} (\hat{p}(x) - W(x)) v^\pi(x)}_{\text{difference in context weights}} \quad (4)$$

$$+ \underbrace{\sum_{x \in \mathcal{X}} W(x) \left( \sum_{x^{(i)}=x} \frac{W^{(i)}}{W(x)W} r^{(i)} - v^\pi(x) \right)}_{\text{weighted IS error in each context}} \quad (5)$$

The first term is a supervised empirical value estimate whose only error is due to the error in the empirical context distribution sampled in the dataset versus the true context distribution. The second term captures the error caused by the difference between context distribution introduced by weights and empirical context distribution in the dataset. The third term computes the difference between the weighted IS estimate of the value of the policy in a specific context $x$ versus its true value $v^\pi(x)$, and then sums this over all contexts.

The second term is of particular interest, because it highlights how the IS estimator of a policy may effectively shift the relative weight on the context space. In the bandit setting (and in the initial starting state distribution for RL), such shifting should not be allowed: the policy may control what actions are taken, but cannot change the initial context distribution. We now show how an algorithm maximizing the importance weighted off-policy estimate can exploit this structure and yield overly inflated estimates (Eq. 5).

**Example 1.** *Consider a contextual bandit problem with $|\mathcal{X}|$ contexts and $|\mathcal{A}|$ actions in each context. For half the contexts $S_p$, the reward is $1$ for one action and zero for others. For the other half of the contexts, $S_n$, the reward is -1 for half the actions, and -5 for the rest. The true distribution over contexts is uniform. The optimal policy would*

*have an expected reward of 0 over the state space. The behavior dataset is drawn from a uniform distribution over contexts and actions. When the sample size $|\mathcal{D}| < |\mathcal{A}|$, we assume there is only one observed positive reward in the dataset. A policy $\pi_o$ that maximizes Eq. 4-5 will select actions that are not present in the dataset for all contexts whose observed actions lead to only zero or negative rewards: let $\mathcal{A}_{s_i} = \{a_i : r(s_i, a_i) \leq 0\}$ then $\pi(s_i) = a_j$ where $a_j \notin \mathcal{A}_{s_i}$. This will yield $W(x) = 0$ on all contexts except for the contexts with observed positive rewards. The resulting IS/SNIS estimator of the value of $\pi_o$ is 1, which is much higher than the optimal value 0. In addition, such a $\pi_o$ is likely to be worse than the optimal policy for any context where $r(s_i, a_i) = -1$, since that is the optimal reward possible for such states $s_i$, and by $\pi$ selecting an unobserved action $a_j$ in that state ($\pi(s_i) = a_j$), the policy $\pi$ may select an action with worse true reward, $r(s_i, a_j) = -5$.*

While this issue can arise in contextual bandits [Swaminathan and Joachims, 2015b], it is even easier for this to occur in sequential RL (Examples 2 and 3 in Appendix A). Intuitively, the issue arises because when estimating the value of a new decision policy, it is acceptable to choose a policy that re-distributes the *weights of actions within an initial context* but not that re-distributes the *weights across initial contexts*, since it is not a function of the actions selected. It is well known that in importance sampling, the expected ratio of the weights should be 1: $\mathbb{E}_{y \sim \mu}[\pi(y)/\mu(y)] = 1$. In contextual policies, we expect that for each initial context $x_0$, the expected weights should also be 1: $\mathbb{E}_{a \sim \mu(a|x_0)}[\pi(a|x_0)/\mu(a|x_0)|x_0] = 1$. However, optimizing for a standard importance sampling objective (such as Eq. 5) does not involve constraints that the empirical expectation of weights given an initial context $\hat{\mathbb{E}}[W_h^{(i)}|x_h^{(i)}]$ (or the weights of $n$-step given initial context $\hat{\mathbb{E}}[W^{(i)}|x_1^{(i)}]$) is still close to one.

Such *propensity overfitting* may seem surprising given that under mild assumptions, which are satisfied here by Assumptions 1 and 2, IS provides an unbiased estimate of a policy's value. Our observations do not contradict this fact: while IS will still provide an unbiased estimate given a policy, policy optimization can exploit the finite sample error and lack of data coverage.

One might hope that existing methods are sufficient to address this challenge. Here we expand on the discussion in the related work to suggest why this is not the case. First, splitting the data into training and selection sets (e.g., [Thomas et al. [2015b], Komorowski et al. [2018]]) is generally insufficient. If some of the performance gains come from systematically avoiding actions taken in initial states with low performance, then it is likely that a similar performance benefit can also arise in the validation set if IS-based estimators are used both in policy selection and later estimation. We will observe experimentally this is true even when the es-

timator or objective involves a variance penalization (Eq. 3). For example, this may occur when only a small set of initial states are avoided, in a way that only mildly impacts the variance and effective sample size, yet results in substantially overly optimistic estimates.

It is also insufficient to shift all rewards to be non-negative. While this voids the benefit of avoiding states if standard IS is used, popular lower-variance IS off-policy estimators like weighted IS are equivariant to any constant shift in rewards. Similarly, doubly robust estimators [Jiang and Li, 2016b, Thomas and Brunskill, 2016] frequently center rewards around estimates of the reward/value outcomes.

*Is constraining to the behavior policy sufficient?* Perhaps the most compelling idea is whether constraining the policy class to actions with some minimal probability under the behavior policy[2]. First note that constraining to the true behavior policy (e.g., Fujimoto et al. [2019], Sachdeva et al. [2020]) can still cause the propensity overfitting problems we described when the dataset is insufficient to cover all non-zero behavior probability actions in all states. In addition, sometimes the behavior policy is itself unknown. Estimating the behavior policy from data and using this in both the IS-based objective and overlap constraints may be more practical given datasets of limited size and when the state space is large. Indeed, semi-parametric theory and past related work in bandits [Narita et al., 2019] and RL [Hanna et al., 2021] have suggested that even if the behavior policy is known, leveraging the estimated behavior policy can yield more accurate offline policy estimates. It is natural to assume such benefits might also translate to improvements for constrained policy learning.

While promising in principle, this approach may be challenging in large state space environments. First, in such settings the maximum number of observed actions in any particular state is almost always one. Assuming there is good reasons to believe that the behavior policy is not actually deterministic, it is necessary to use some function approximation method to estimate the behavior policy [Hanna et al., 2021], which may be a deep neural network or non-parametric methods like k-nearest neighbors [Raghu et al., 2018]. Unfortunately, as we will demonstrate in our experiments, we find that such approximators may sometimes be sufficient to accurately predict the behavior policy for a given state, but do not seem to be as beneficial when used to constrain the targeted policy class. Such estimates may overestimate (or underestimate) the probability of taking alternate actions in some states, and therefore enable both context avoidance or be too conservative in their policies. Figure 1 illustrates that our method can provide quite different constraints on the policy class than using constraints on the estimated empirical behavior policy, shown for a patient in the MIMIC

---

[2]A minimal requirement for consistent estimation of a target policy using IS is that there is overlap between the behavior policy and target, which we also assume.

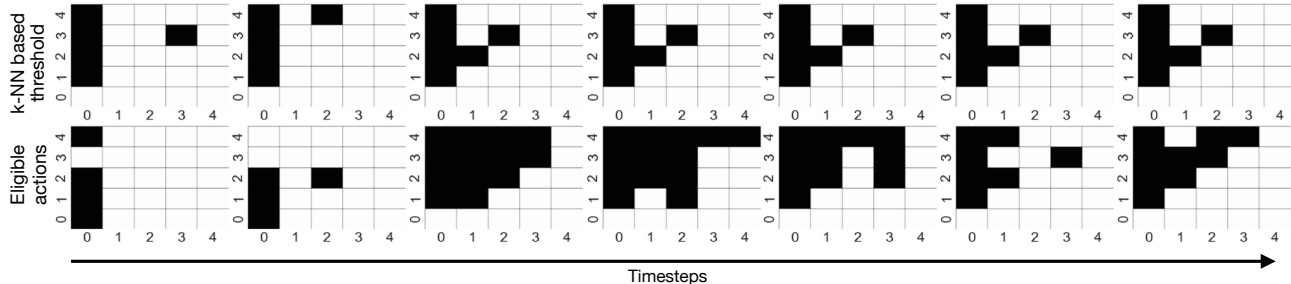

Figure 1: Different policy class constraints in MIMIC sepsis data. The top shows in black which of the 25 actions pass a constraint on having minimal probability over an estimated behavior policy, $\mu_{KNN}(a|s) > b$, for a sequence of patient states. The bottom shows POELA's eligible actions for the same states. This illustrates that even constraints given the empirical estimated behavior policy can be very different than POELA's eligible actions. Note sometimes POELA is more conservative, and other times the estimated behavior policy constraint is more conservative.

sepsis dataset.

## 5 POLICY OPTIMIZATION WITH ELIGIBLE ACTIONS

We have described that in IS-based estimators, because all weight can be placed on unobserved actions for certain initial contexts, the empirical conditional expectation of weights given an initial context $\hat{\mathbb{E}}[W|x]$ can be zero for such contexts, instead of 1. To address this, one possibility is to constrain the conditional expectation of weights given a context to be 1 or lower bounded. However doing so in infinite/continuous context spaces is subtle: each context likely only appears once in the dataset, and requiring $\hat{\mathbb{E}}[W|x] = 1$ would be equivalent to only allowing a policy that exactly matches the observed logged actions.

We now propose a slight relaxation of the above proposal. Absence constraints, IS-estimator-based policy learning can place a large weight on unobserved actions in the dataset, for which our reward uncertainty is high. The recent line of pessimism under uncertainty for model and value based MDP offline RL (e.g., Liu et al. [2020], Yu et al. [2020]) explicitly accounts for such statistical uncertainty through constraining or penalizing actions and/or states and actions for which there are limited observed data. Our approach is similar but designed to address these issues in settings that may not be Markov.

Specifically, for policy learning, we create local constraints on the eligible policy class. For each context $x$, dataset $\mathcal{D}$ and a given threshold $\delta$, the *eligible action* set $A(x; \mathcal{D}, \delta)$ is:

$$A_h(x; \mathcal{D}, \delta) = \{a_h : \exists (x_h, a_h) \in \mathcal{D} \ s.t. \ \text{dist}(x, x_h) \leq \delta\}.$$

This allows only the action that was taken for a given context, or actions taken in contexts within a given distance $\delta$ of the observed contexts. Note that in large or continuous action spaces, the resulting allowed actions per context may

be quite different than policy search methods that places thresholds using the behavior policy probability [Futoma et al., 2020] as we will observe empirically (cf. Figure 1). Intuitively, our eligible action constraint constraints the policy class to *observed* actions taken in the current or nearby[3] states, but behavior policy thresholds can allow actions that could be taken at that state, even if no such action was taken there, nor at any nearby state. We will shortly prove that our eligible action set is sufficient to lower bound the empirical conditional expectation of weights per observed context. This ameliorates the propensity overfitting problem, and we will shortly show that empirically this can yield significant benefits.

Policy learning can be done by finding the best policy that satisfies the eligible actions constraints given the dataset:

$$\arg\max_{\pi \in \Pi} J(\pi; \mathcal{D})$$
$$s.t. \ \forall i, h \ \sum_{a \in A_h(x_h^{(i)}; \mathcal{D}, \delta)} \pi(a|x_h^{(i)}) = 1. \quad (6)$$

$J(\pi; \mathcal{D})$ can be any objective function such as $\hat{v}_{\text{IS}}$ or $\hat{v}_{\text{SNTIS}}$. We now present our POELA (Policy Optimization with ELigible Actions) algorithm (Algorithm 1) that implements the learning objective in Equation 6. We use the counterfactual risk minimization objective function (Eq. 3) as the $J(\pi; \mathcal{D})$, where the estimator $\widehat{\text{Var}}(\hat{v}_{\text{SNTIS}})$ is constructed using the Normal approximation in [Owen, 2013, Equation 9.9]: $\widehat{\text{Var}}(\hat{v}_{\text{SNTIS}}) = \frac{\sum_{i=1}^{n}(r^{(i)} - \hat{v}_{\text{SNTIS}})^2(\min\{W^{(i)}, M\})^2}{(\sum_{i=1}^{n} \min\{W^{(i)}, M\})^2}$.

After each gradient step, we enforce the policy to satisfy

---

[3]In this work, we use the Euclidean distance on the state space, but for very high dimensional state spaces, it would likely be beneficial to compute distances leveraging representation learning RL work (e.g., Zhang et al. [2021]). We also considered other approaches to defining a "neighborhood" of a given context: k-nearest neighbors. The problem was that this could include samples that were very far away in context space, and so it could appear like there was local similarity and coverage of the policy, but in reality no nearby states had taken a similar action.

**Algorithm 1** Policy Optimization with ELigible Actions (POELA).

1: **Input:** $\mathcal{D}$, $\Pi_\theta$, sphere radius $\delta$, IS truncation $M$, CRM coefficient $\lambda$, learning rate $\alpha$
2: **Output:** $\widehat{\pi}_\theta$
3: Initialize $\theta_0$
4: **for** $t = 0, 1$ **until convergence do**
5:
$$\widehat{\pi}_{\theta_t}(a|x) := \frac{\mathbb{1}\{a \in A_h(x; \mathcal{D}, \delta)\}}{\sum_a \mathbb{1}\{a \in A_h(x; \mathcal{D}, \delta)\}\pi_{\theta_t}(a|x)}\pi_{\theta_t}(a|x)$$
6:
$$\theta_{t+1} \leftarrow \theta_t + \alpha \nabla_\theta \left( \hat{v}_{\text{SNTIS}}(\widehat{\pi}_{\theta_t}) - \lambda \sqrt{\widehat{\text{Var}}\left(\hat{v}_{\text{SNTIS}}(\widehat{\pi}_{\theta_t})\right)} \right)$$
7: **end for**

---

the eligible action constraints by re-normalizing the output probability on $A_h(x; \mathcal{D}, \delta)$ for $x \in \mathcal{D}$. The eligible action set for each training sample is static and can be stored to reduce computational cost. In the experiments, we use the Euclidean distance over nearby states at any time index.

## 6 ANALYSIS

In this section we formalize the benefit of the eligible action constraints, and also prove consistency guarantees on the resulting value estimates used in the objective, Equation 6.

Eligible action constraints were introduced to help alleviate propensity overfitting, which can be characterized by the expected empirical sum of propensity weights for a particular context $x$ being much lower than 1, or even $\hat{\mathbb{E}}[\mathbf{W}|\mathbf{x}] = 0$. We now show that any policy that only selects actions in eligible action sets will ensure that the empirical sum of weights in any hypersphere of any context can be lower bounded, as desired. As mentioned before, for very large state spaces, where only a single action is observed for each observed state in the dataset, the only way to ensure that $\hat{\mathbb{E}}[\mathbf{W}|\mathbf{x}] = 1$ is to reduce the policy to the observed actions. Intuitively, the guarantee we provide here is reasonable when local smoothness is present and a soft form of state aggregation is tenable, ensuring that the empirical expected sum of weights over nearby states is lower bounded.

To do so, we first introduce an assumption about the target policy's smoothness in the context space.

**Assumption 3** (*L*-Lipschitz policy). $\forall \pi \in \Pi$, $\|\pi(a|x) - \pi(a|x')\| \leq L\text{dist}(x, x')$.

That is, nearby contexts have similar actions [Berkenkamp et al., 2017, Wang et al., 2019] which ensures that we have some minimal weight support in a small neighborhood. If different policies have different smoothness, the Lipschitz constant can be taken to be the max over the policy set.

Under this assumption and the former assumptions, we can show the following. Proofs are provided in Appendix B:

**Theorem 1.** $\forall x_h^{(i)}$, $\mathcal{B}(x_h^{(i)}, \delta) := \{x : \text{dist}(x, x_h^{(i)}) \leq \delta\}$, $\sum_{x_h^{(j)} \in \mathcal{B}(x_h^{(i)}, \delta)} W_h^{(j)} \geq 1 - \delta L|\mathcal{A}|$.

Given the likelihood ratio is lower bounded, we can further show that the self-normalized truncated weights are also lower bounded in the one-step settings.

**Corollary 1.** *For* $H = 1$, $\sum_{x_1^{(j)} \in \mathcal{B}(x_1^{(i)}, \delta)} \frac{\max\{W^{(i)}, M\}}{\sum_{i=1}^n \max\{W^{(i)}, M\}} \geq \frac{1 - \delta L|\mathcal{A}|}{nM}$ *for* $M > 1$.

In the $n$-step sequential setting, it is necessary to have the 1-step weights be greater than zero in order to have $n$-step weights greater than zero.

**Proposition 1.** *For any* $x$, $\delta$, $\mathbb{E}[W_{1:h}^{(i)}|x_h^{(i)} \in \mathcal{B}(x, \delta)] = \mathbb{E}[W_{1:h-1}^{(i)}|x_h^{(i)} \in \mathcal{B}(x, \delta)]\mathbb{E}[W_h^{(i)}|x_h^{(i)} \in \mathcal{B}(x, \delta)]$. $\hat{\mathbb{E}}[W_{1:h}^{(i)}|x_h^{(i)} \in \mathcal{B}(x, \delta)] = \hat{\mathbb{E}}[W_{1:h-1}^{(i)}|x_h^{(i)} \in \mathcal{B}(x, \delta)]\hat{\mathbb{E}}[W_h^{(i)}|x_h^{(i)} \in \mathcal{B}(x, \delta)]$.

The action eligibility local constraints provide a conservative pessimism-like constraint on the policy class, ensuring that the policy does not take actions which have not been tried in nearby states (and therefore for which the potential outcomes are unknown). As we will see shortly, this will yield more stable and beneficial performance in our simulations. An additional desirable property is that asymptotically, POELA relaxes to unconstrained policy learning.

**Theorem 2.** *(Contains all overlapping policies). For a fixed* $\delta$, *for any* $x$, $A_h(x; \mathcal{D}, \delta) \rightarrow \{a : \mu(a|x) > 0\}$ *as* $n \rightarrow \infty$ *with probability 1. Therefore asymptotically the policy class will contain all* $\pi$ *satisfying the overlap assumption.*

**Theorem 3.** *Let* $J(\pi, \mathcal{D})$ *be the objective* [4] *in Equation 3, the truncation threshold* $M$ *as a function satisfies* $M \rightarrow \infty$ *and* $M/n \rightarrow 0$ *as* $n \rightarrow \infty$, *and* $|\Pi| < \infty$, *then* $v^{\hat{\pi}_{\mathcal{D}, J}} \rightarrow \max_{\pi \in \Pi} v^\pi$ *in probability.*

## 7 EXPERIMENTS

We now compare POELA with several prior methods for offline RL. Perhaps the most relevant work in avoiding overfitting when using importance sampling is norm-POEM [Swaminathan and Joachims, 2015b]. For it to be suitable for sequential decision settings, we use a neural network policy class and refer to the resulting algorithm as PO-CRM. A second baseline PO-$\mu$ constrains the policy class to only include policies which take actions with a sufficient probability under the behavior policy $\mu(a|s)$ [Futoma et al.,

---

[4]Other consistent estimators $J$ can also be shown to satisfy this property, such as IS and self-normalized IS.

| | Algorithms | POELA | PO-$\mu$ | PO-CRM | BCQ | PQL | 9-mon |
|---|---|---|---|---|---|---|---|
| Non-MDP | Test $v^\pi$ | $95.92 \pm 1.68$ | $76.99 \pm 13.80$ | $77.32 \pm 14.55$ | $13.60 \pm 0.15$ | $19.64 \pm 5.71$ | $68.12$ |
| | $\hat{v}_{\text{SNTIS}} - v^\pi$ | $-1.28 \pm 1.93$ | $16.07 \pm 13.55$ | $15.71 \pm 14.30$ | $80.54 \pm 1.42$ | $74.48 \pm 6.23$ | $-$ |
| MDP | Test $v^\pi$ | $89.53 \pm 1.32$ | $69.18 \pm 10.17$ | $63.27 \pm 13.27$ | $82.68 \pm 15.19$ | $99.98 \pm 0.38$ | $68.12$ |
| | $\hat{v}_{\text{SNTIS}} - v^\pi$ | $5.12 \pm 2.01$ | $24.92 \pm 9.71$ | $30.82 \pm 12.59$ | $14.76 \pm 14.89$ | $-2.65 \pm 1.76$ | $-$ |

Table 1: LGG Tumor Growth Inhibition simulator. Test $v^\pi$ (1000 rollouts in the simulator) and $\hat{v}_{\text{SNTIS}} - v^\pi$ (amount of overfitting of the learned policy) with $\hat{v}_{\text{SNTIS}}$ on the validation set. Average across 5 runs with standard error reported.

2020]. We also compare with recent deep value-based MDP methods in batch RL: BCQ [Fujimoto et al., 2019] and PQL [Liu et al., 2020]. For all algorithms, we use a feed-forward neural network for the relevant policy and/or value function approximators. We report the test performance of the selected policy either through online Monte-Carlo estimation if a simulator is available, or using SNTIS estimates on a held out test set. Full details are provided in Appendix C.

### 7.1 TUMOR INHIBITION SIMULATOR

The Tumor Growth Inhibition (TGI) simulator [Ribba et al., 2012] describes low-grade gliomas (LGG) growth kinetics in response to chemotherapy in a horizon of 30 steps (months), with a non-Markov context and a binary action of drug dosage [Yauney and Shah, 2018]. The reward is an immediate penalty proportional to the drug concentration, and a delayed reward of the decrease in mean tumor diameter. The behavior policy selects from a fixed dosing schedule of 9 months (the median duration from Peyre et al. [2010]) with 70% probability and else selects actions at random.

In this experiment, the behavior policy can only take values in $\{0.15, 0.85\}$. This means that constraining the policy class to have a minimal probability under $\mu(a|s)$, as in baseline PO-$\mu$, is only a non-trivial constraint for thresholds greater than $0.15$: this generates a single potential target policy, which is the deterministic fixed-dosage part of behavior policy. We include this as 9-mon (short for 9 month dosing) in Table 1. The training and validation sets both have 1000 episodes. We repeat the experiment 5 times with 5 different train and validation sets. Policy values are normalized between 0 (uniform random) and 100 (best policy from online RL). As shown in Table 1 (Non-MDP rows), POELA achieves the highest test value as well as smaller variability compared with the baselines.

**Does POELA reduce overfitting?** Examining the difference between $\hat{v}_{\text{SNTIS}}$ on the validation set and the online test value, we observe that most algorithms result in a policy whose value is a significant overestimate of its true performance (cf. Table 1, $\hat{v}_{\text{SNTIS}} - v^\pi$). In contrast, POELA yields a policy whose value is much more accurately estimated and performs better. Experiments with final policies selected during training based on SNTIS estimates on the validation set suggest the same conclusion (cf. Table 12 in Appendix D.2).

**Performance comparison in a MDP environment.** We also repeat the experiment with an MDP modification of the simulator, including an immediate Markovian reward and additional features for a Markovian state space. Note that we expect BCQ and PQL to do very well: both are designed to avoid overfitting in offline MDP learning and in particular PQL uses a pessimism under uncertainty approach to penalize policies that put weight on state-action pairs with little support. Although POELA makes no Markov assumptions, it ponly erforms on average slightly worse than the two conservative MDP methods but still outperforms BCQ. POELA also substantially outperforms other policy classes.

### 7.2 MIMIC III SEPSIS ICU DATA

Next, we apply our method in a real-world example of learning policies for sepsis treatment in medical intensive care units (ICU). We used an extracted cohort [Komorowski et al., 2018] of patients fulfilling the sepsis-3 criteria from the MIMIC III data set [Johnson et al., 2016] and obtained a dataset of 14971 patients, 44 context features, 25 actions and a 20 step maximum horizon. Full details are in the Appendix C.2. We hold out 20% of data for validation and 20% of data for the final test. Treatment logs do not include the probabilities of clinicians' actions. Instead, as suggested by prior work [Raghu et al., 2018], we estimate the probabilities of the behavior clinicians' policy by $k$-NN with $k = 100$. To ensure overlap, for all policy optimization algorithms we allow $\pi(a|s) > 0$ only if $\hat{\mu}(a|s) > 0$. Using SNTIS to evaluate the performance on a test set is appealing because it makes little assumptions on the underlying domain. But if only a few test behavior policy trajectories match a test policy, the resulting value estimate is likely unreliable. We measure the amount of overlap between the test set and a desired policy by the effective sample size (ESS) [Owen, 2013]. Only policies with an ESS of at least 200 on the validation set are considered.[5] Similar to prior work [Thomas et al., 2015a], in addition to the SNTIS estimator on the test set, we also report a 95% upper and lower bound from bias-corrected and accelerated (BCa) bootstrap. The clinician's

---

[5]The variance penalty may not ensure that the ESS is large, because it is only a soft penalty rather than a constraint that ensures a minimum ESS.

| Method | POELA | PO-$\hat{\mu}$ | PO-CRM | BCQ | PQL | Clinician |
|---|---|---|---|---|---|---|
| Test SNTIS | 92.32 (90.87) | 90.21 | 86.89 | 25.62 | 27.04 | 81.10 |
| 95% BCa UB | 95.83 (92.94) | 93.27 | 89.68 | 41.93 | 42.45 | 82.19 |
| 95% BCa LB | 90.91 (87.22) | 87.19 | 83.50 | 7.93 | 13.43 | 79.80 |
| Test ESS | 437.03 (396.71) | 297.84 | 289.10 | 206.63 | 217.54 | 2995 |

Table 2: MIMIC III sepsis dataset. Test evaluation, $(0.05, 0.95)$ BCa bootstrap interval, and ESS. The value of POELA without a CRM variance penalty is in parentheses.

| Method | POELA | PO-$\hat{\mu}$ | PO-CRM | BCQ | PQL |
|---|---|---|---|---|---|
| Test SNTIS | 86.42 (85.26) | 84.39 | 79.71 | 32.83 | 34.69 |
| 95% BCa UB | 91.68 (90.32) | 87.74 | 89.33 | 53.50 | 52.15 |
| 95% BCa LB | 79.71 (77.15) | 80.01 | 65.01 | 11.87 | 17.60 |
| Test ESS | 310.23 (287.39) | 244.97 | 224.92 | 207.12 | 223.03 |

Table 3: Idem except using behavior policy $\hat{\mu} = $ BC.

| Method | POELA | PO-$\hat{\mu}$ | PO-CRM | BCQ | PQL |
|---|---|---|---|---|---|
| Test SNTIS | 88.83 (88.31) | 87.97 | 85.33 | 33.21 | 41.66 |
| 95% BCa UB | 93.23 (94.04) | 91.17 | 89.44 | 63.17 | 57.99 |
| 95% BCa LB | 83.43 (80.02) | 82.00 | 78.43 | 12.23 | 14.76 |
| Test ESS | 379.18 (265.36) | 220.74 | 236.78 | 203.89 | 224.33 |

Table 4: Idem except using behavior policy $\hat{\mu} = $ BCRNN.

column is the test dataset rewards and sample size.

Table 2 shows POELA is the best on all metrics, achieving the highest evaluation on the test set, the highest upper and lower bounds, and the highest ESS. We also show that POELA's test performance without its variance penalty is worse than using it but is still higher than the baseline algorithms. In Appendix E.3, we further detail the differences in the methods under the prism of ESS and performance. We also demonstrate in Appendix E.4 that POELA takes actions which more closely match the clinicians' actions for patients with initially high logged SOFA scores (measuring organ failure) in the test dataset in comparison to other baselines, suggesting that propensity overfitting may be occurring more in other methods. Finally, Figure 1 illustrates different actions constraints considered in this paper, using a sample trajectory in the test set as an example, where the 25 actions are depicted in 5x5 grids.

### 7.3 BEHAVIOR POLICY ESTIMATION

We now explore further if constraining policies to be close to the empirical behavior policy may produce similar benefits, and whether this depends on the function approximator used. We consider two additional function approximators: (1) learning a deep neural network representation of the behavior policy using Behavior Cloning (BC), an imitation learning approach [Pomerleau, 1991] and (2) training a recurrent neural network behavior representation using BCRNN, a variant of BC with a RNN as the policy network. BCRNN can learn temporal dependencies, which can be helpful. More details are included in Appendix C.3.

Results in the MIMIC III sepsis dataset are shown in Tables 3 and 4. Results in the Tumor simulator are included in Tables 10 and 11. Overall, this behavior policy modelling modification impacts all methods, but POELA still outperforms other baselines. We note that BCQ and PQL benefit

from these alternate behavior policy approximators, while policy-based methods suffer from it in the Non-MDP setting. Comparing the benefits of using BC versus BCRNN, BCRNN behavior policy approximators in the non-Markov settings generally helps, as expected. We report additional results where best policies are selected from checkpoints during training based on SNTIS estimates in Tables 13 and 14 (tumor) and in Tables 16 and 17 (sepsis). In this application-driven selection procedure, POELA still yields higher test values.

### 7.4 EXPERIMENT WITH CONTINUOUS STATE SPACE

In the next experiment, we use the OpenAI Gym environment [Brockman et al., 2016] CartPole. We also apply our method to a non-Markov modification of the environment. More details about this experiment in Appendix F. In these experiments, only policies with an ESS of at least 30 on the validation set are considered. Because of space constraints, the full results are provided in Appendix F. Tables 20 to 23 show the results. We observe that in both MDP and Non-MDP settings, POELA provides improved performance over other methods. We also provide some results on D4RL [Fu et al., 2020] datasets in Appendix G. These results show that relying on observed data to decide on action eligibility can be beneficial for learning from the relatively few number of trajectories collected by the behavior policy in a continuous state space.

## 8 DISCUSSION & CONCLUSION

A natural question is whether POELA, in addition to its overall improved performance, reduces context avoidance/propensity overfitting in practice. In Appendix C.4, we find that POELA generally puts more weight on ini-

tial states with low observed outcomes than other IS policy optimization methods, suggesting that it addresses the motivating problem. POELA also does not seem highly sensitive to the threshold used in the eligible action constraint (cf. Appendix C.5) although middle ranges are more effective.

An alternative to constrained optimization is a soft penalty based on the proportion of contexts that are avoided through selecting alternative actions. This idea was previously proposed for contextual bandits [Sachdeva et al., 2020]. This is challenging to approximate in the RL setting, where deficiency can occur at any steps in a trajectory: exploring this is an interesting area for future work. Another interesting direction is to adapt the solution found in Joachims et al. [2018] when using a minibatch biases the SNTIS estimate.

To conclude, we identify a new overfitting problem arising when using IS as part of an offline policy learning objective. To address this, we constrain the policy class to only consider logged actions taken by nearby states. This can be viewed as a pessimism constraint similar to the one used in MDP offline policy learning, but developed for a non-Markov, direct policy search setting. Our approach yields strong performance relative to state-of-the-art approaches in a tumor growth simulator, a real-world dataset on ICU sepsis treatment and in classic continuous control with few demonstrations. POELA may be particularly useful for many applied settings such as healthcare, education and customer interactions, which have a short/medium length decision horizon, but are unlikely to be Markov in the observed per-step variables. Leveraging constraints on an empirical behavior policy was not as helpful, but an interesting direction for future work is whether other ways of learning such behavior policy might yield additional benefits to our locally constrained approach.

## Acknowledgements

Research reported in this paper was sponsored in part by NSF grant #2112926 and the DEVCOM Army Research Laboratory under Cooperative Agreement W911NF-17-2-0196 (ARL IoBT CRA). The views and conclusions contained in this document are those of the authors and should not be interpreted as representing the official policies, either expressed or implied, of the Army Research Laboratory or the U.S. Government. The U.S. Government is authorized to reproduce and distribute reprints for Government purposes notwithstanding any copyright notation herein.

The first and last authors acknowledge the Simons Institute for the Theory of Computing as this research is initiated when the two authors were a visiting student and long-term participant, respectively, at the Theory of Reinforcement Learning program of the Simons Institute.

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
