# OpenReview forum: "Offline Policy Optimization with Eligible Actions"
_auai.org/UAI/2022/Conference — UAI 2022 Poster_

### Official Review · Reviewer_vL39 · 2022-04-12

**Q2(1) Originality/Novelty:** 2
**Q2(2) Significance/Impact:** 3
**Q2(3) Correctness/Technical Quality:** 3
**Q2(6) Clarity Of Writing:** 3
**Q6 Overall Score:** 5
**Q8 Confidence In Your Score:** 4

**Q1 Summary And Contributions:**

This paper studies the offline policy optimization problem using the pessimism-based learning in the contextual bandit (non-Markovian) setting. This paper proposes an algorithm to help alleviate the issue of propensity overfitting. Standard asymptotic guarantees that offline policy evaluation papers make via Slutsky’s theorem are provided and complemented with experimental results.

**Q10 Ethical Concerns (Optional):**

No.

**Q2 Assessment Of The Paper:**

More detailed information regarding each of these aspects is given below:

**Q2(4) Quality Of Experiments (Optional):**

3: Good: The experimental evaluation is adequate, and the results convincingly support the main claims.

**Q2(5) Reproducibility:**

2: Fair: Key resources (e.g., proofs, code, data) are unavailable but key details (e.g., proof sketches, experimental setup) are sufficiently well-described for an expert to confidently reproduce the main results.

**Q3 Main Strengths:**


+ The paper is well written and well structured. Especially Section 4 on the negative results of different popular alternative solutions are clearly mentioned.

+ Authors provides extensive evaluation on 3 different environments comparing with 3+ benchmarks. One of the environments is healthcare related which, in my opinion, helps broaden the paper's impact.


**Q4 Main Weakness:**

Please refer to my response to Q5.

**Q5 Detailed Comments To The Authors:**

(1) The authors provide some statistical guarantees with similar works, but not competitive enough when compared to the finite sample guarantees in Switch-Doubly-Robust [c1], DR-optimistic shrinkage [c2], etc.

(2) Assumption 1 and 2 are common in the literature. But details on Assumption 3 are missing in this paper. The assumption that “nearby contexts have similar actions,” seems reasonable, but having the SAME Lipschitz constant for all policies and for all actions is rather unrealistic! Please justify this including any past literature which uses this assumption.

[c1] Wang, Yu-Xiang, Alekh Agarwal, and Miroslav Dudık. "Optimal and adaptive off-policy evaluation in contextual bandits." International Conference on Machine Learning. PMLR, 2017.

[c2] Su, Yi, et al. "Doubly robust off-policy evaluation with shrinkage." International Conference on Machine Learning. PMLR, 2020.

-------------------- General Remarks --------------------

(1) P3C1-paragraph 1: “penalization do not directly the problem with avoid” -> “penalization do not directly solve the problem with avoid”
(2) P4C1-paragraph 2 end: Typo with extra $x_1^{(i)}$ condition in the expectation.
(3) One subset study on using different distance metrics for the contexts is useful. Please add comments on why the Euclidean metric might be preferred regardless of the application.
(4) Table 1, $\hat{v}-v^\pi$ rows. Is the $v^\pi$ set common across all the algorithms?


**Q7 Justification For Your Score:**

Please see my responses to Q4 and Q5 above.

**Q9 Complying With Reviewing Instructions:**

1: Yes.

---

### Official Review · Reviewer_VTn3 · 2022-04-18

**Q2(1) Originality/Novelty:** 2
**Q2(2) Significance/Impact:** 2
**Q2(3) Correctness/Technical Quality:** 3
**Q2(6) Clarity Of Writing:** 3
**Q6 Overall Score:** 4
**Q8 Confidence In Your Score:** 4

**Q1 Summary And Contributions:**

This paper considers offline policy optimization using importance weighted estimation and identifies an overfitting phenomenon associated with the importance weights. The paper presents an algorithm to handle this issue, and presents theoretical support and experimental results that validate this approach.

**Q2 Assessment Of The Paper:**

More detailed information regarding each of these aspects is given below:

**Q2(4) Quality Of Experiments (Optional):**

2: Fair: The experimental evaluation is weak: important baselines are missing, or the results do not adequately support the main claims.

**Q2(5) Reproducibility:**

2: Fair: Key resources (e.g., proofs, code, data) are unavailable but key details (e.g., proof sketches, experimental setup) are sufficiently well-described for an expert to confidently reproduce the main results.

**Q3 Main Strengths:**

The paper identifies an overfitting phenomenon when using importance weights in offline RL - extending what was known in the offline contextual bandit literature (Swaminathan and Joachims 2015), which is a clear contribution. The fix presented has been developed in the contextual bandit literature, and is extended to the RL setting. This has been complimented with theoretical support and experimental results highlighting the effectiveness of the proposed approach.

**Q4 Main Weakness:**

Experimental results wise, the contributions are limited owing to:
(a) not considering standard benchmark D4RL tasks to consider impact of various algorithm building blocks when varying the logging policy used to collect data, and,
(b) in order to compare this paper's result against a suite of existing offline RL methods.

**Q5 Detailed Comments To The Authors:**

(Missing) References:
Yao Liu, Adith Swaminathan, Alekh Agarwal, Emma Brunskill, "Off-Policy Policy Gradient with State Distribution Correction", UAI 2019


**Q7 Justification For Your Score:**

See Q3/Q4.

**Q9 Complying With Reviewing Instructions:**

1: Yes.

---

### Official Review · Reviewer_23ZV · 2022-04-18

**Q2(1) Originality/Novelty:** 3
**Q2(2) Significance/Impact:** 3
**Q2(3) Correctness/Technical Quality:** 3
**Q2(6) Clarity Of Writing:** 3
**Q6 Overall Score:** 7
**Q8 Confidence In Your Score:** 3

**Q1 Summary And Contributions:**

This paper addresses the issue of propensity overfitting in offline contextual bandits and RL settings by restricting to a policy class that only allows (context, action) pairs that have enough support in the training logs.


**Q2 Assessment Of The Paper:**

More detailed information regarding each of these aspects is given below:

**Q2(4) Quality Of Experiments (Optional):**

3: Good: The experimental evaluation is adequate, and the results convincingly support the main claims.

**Q2(5) Reproducibility:**

3: Good: Key resources (e.g., proofs, code, data) are available and key details (e.g., proofs, experimental setup) are sufficiently well-described for competent researchers to confidently reproduce the main results.

**Q3 Main Strengths:**

Reasonably clear presentation of the method.  Nice selection of experimental settings.


**Q4 Main Weakness:**

Could use more explanation of how much POELA deviates from the behavior policy in performance. Also, for the settings where you have simulators, how different are the SNTIS estimators from the 'ground truth' value, to help get a sense for how reliable using SNTIS on validation and test is for settings without a simulator.


**Q5 Detailed Comments To The Authors:**

- Typo in the first definition of the IS estimator, where you don't use subscript t. Similarly in equation (1).
- "When the sample size |D| < |A|, we can assume there is only one observed positive reward in the dataset."  This seems strange.
- Example (1) refers to maximizing Equation 5, but that's the last term in the equation started with (4).  I think the numbering should be fixed?
- The idea in Example 1 was one of the important observations in Swaminathan and Joachims' 2015 paper on self-normalized estimator.  Maybe should give a reference to this or something earlier?
- "Intuitively, the issue arises because in estimating the value of a new decision policy, it is acceptable to choose a policy that re-distributes the weights of actions within an initial context but not re-distribute the weights across initial contexts, since it is not a function of the actions selected."  I'm having trouble understanding this sentence...
- Did you consider other approaches to defining a "neighborhood" of a given context, besides a fixed radius delta?  e.g. k-nearest neighbors?
- One interesting aspect of the self-normalized IS estimator (as well as the variance penalty) from Swaminathan and Joachims 2015b
is that using a minibatch biases the estimate.  A followup 2018 paper (Deep learning with logged bandit feedback) proposed a workaround.  Did you give this issue any consideration in your optimization?
- In Table 1, it seems like we should have a column for the performance of the behavior policy. Your whole method is based on not deviating too far from the behavior policy (in a certain sense), so it's of interest how far performance of POELA deviates from the behavior policy.
- You say in footnote 5 "the variance penalty may not ensure that the ESS is large".  They do seem to be getting at the same idea.  Is there an example or an intuitive explanation for what the essential difference is? If so, this might be useful to incorporate.


**Q7 Justification For Your Score:**

I think the method is a nice contribution to a problem of broad interest that is nicely evaluated.


**Q9 Complying With Reviewing Instructions:**

1: Yes.

---

### Official Review · Reviewer_mWzt · 2022-04-19

**Q2(1) Originality/Novelty:** 1
**Q2(2) Significance/Impact:** 2
**Q2(3) Correctness/Technical Quality:** 2
**Q2(6) Clarity Of Writing:** 1
**Q6 Overall Score:** 3
**Q8 Confidence In Your Score:** 3

**Q1 Summary And Contributions:**

The paper addresses offline policy optimization, where the authors identify an important overfitting phenomenon in optimizing the importance weighted return and propose an algorithm to avoid this overfitting.  A few basic theoretical analysis of their approach is provided along with some good experimental evaluation which includes healthcare-inspired simulator and a continuous control task.


**Q2 Assessment Of The Paper:**

More detailed information regarding each of these aspects is given below:

**Q2(4) Quality Of Experiments (Optional):**

3: Good: The experimental evaluation is adequate, and the results convincingly support the main claims.

**Q2(5) Reproducibility:**

2: Fair: Key resources (e.g., proofs, code, data) are unavailable but key details (e.g., proof sketches, experimental setup) are sufficiently well-described for an expert to confidently reproduce the main results.

**Q3 Main Strengths:**

I think the idea of the paper is good. The empirical evaluation seems to supplement this claim. The experimental evaluation is also decent enough.

**Q4 Main Weakness:**

The exposition of the paper is very poor: I feel that the paper has some potential, however I found that the paper in this current form is too hard to follow. The authors have to rework the current exposition of the paper, recomb it and make it into a more readable format. I was struggling throughout to understand the details including the problem statement, the logic, and even the theoretical results. I feel that some theoretical results are being added just to look good, rather than adding any proper value to the paper. It is very important that one has to provide intuitive meaning to the theoretical results. Without that one cannot judge how the theoretical results quantify the algorithm/approach.


**Q5 Detailed Comments To The Authors:**

I could not understand, what is the real problem being addressed in the paper.  What I could minimally understand from para 2, Section 2 is that the problem addressed is to find the best policy (from a collection of policies) which best approximates a given sample trajectory generated using a behavior policy. If this is the case, then what is the motivation for such a problem.  Why should one want to find the best policy to represent a sample trajectory? Can you provide some real examples?  Even the following phrase from Section 2 para 2 creates more confusion: ", and we aim to find a policy π in a policy class Π with the highest value", does not make sense to me. What does it really mean to say the highest value? Can you provide some formal way to express the statement in the manuscript? Without sufficient details about the problem itself,  it is really painful to understand the real contribution of the paper itself.

Also in the following para, what does this statement mean? "Policy gradient and optimization approaches do not rely on a Markov assumption on the underlying domain, and have had some encouraging success in offline RL".  How is it possible to say that the Policy gradient does not require any Markov assumption? What are the references you can provide to justify this statement?


Regarding Assumption 1: Are you assuming that the action space and state space are finite. If so, then this assumption is directly assumed. If not, then this assumption is too strong. Because in the case of continuous state and action spaces, one cannot assume that the target and behaviour are close to each other.

In Section 4, para 2, what is v_{\pi}(x) ? Is it single-step expected reward?
In Page 4, Left column, para 2:  The following sentence: " It is well known that in importance sampling, the expected ratio of the weights should be 1". It is not WELL KNOWN, it is actually TRIVIAL.

Also in the following sentence: ". In contextual policies, we expect that for each initial context x0, the expected weights should also be 1:". Again, this expectation will also be 1.  I could not understand, why one should "expect" this condition.

In Algorithm 1, it is computationally very exhaustive to compute to Indicator random variable in Step 5 of the algorithm. This is because one has to traverse the whole data set again and again to find  \delta-close actions from the data set D.

Again in Algorithm 1, how does one ensure that the gradient indeed exists in step 6 of the algorithm. I do not see any structural assumption being made in the paper before which justifies this requirement. These are critical factors one should provide in the paper to justify the credibility of the algorithm.

What is the guarantee that Algorithm 1 itself converges?

What is the variable M in the last para of Page 5 ? Also in the same para, what is v_{IS}?

What is the meaning of "overlap assumption" in Theorem 2?

From Theorem 1, it seems that the action space is finite since the result has |A| in the lower bound. But I don't see any statement in the paper, which assumes that the action space is indeed discrete and finite.

Also in Theorem 3, I don't see where the variables used in the statement of the theorem are defined? For example, what is M ? what is v^{\pi D, J}?

Assumption 3 does not even make sense. Note that policy \pi is a function of action condition on the state. How can one assume the Lipschitz continuity of state considering that action is involved? There is something amiss here. What is the notion of "dist" in Assumption 3? What is the definition of norm || used in Assumption 3?


**Q7 Justification For Your Score:**

It is indeed really hard to understand the real details of the paper as so much information pertaining to the problem itself and the proposed algorithm is missing. Maybe I am wrong in judging the paper itself. But I think I was kind of constrained to appreciate the work involved in the paper due to the poor exposition. I strongly believe that in this current form, the paper is not fit to be published.

**Q9 Complying With Reviewing Instructions:**

1: Yes.

---

### Decision · Program_Chairs · 2022-05-15

**Decision:**

Accept (Poster)

**Comment:**

Meta Review: The reviewers have quite divergent opinions towards this paper. Generally the reviewers recognize that the problem is well studied with extensive experiments and proper theoretical justifications. While there still exist several issues, the authors have provided additional results and clarifications, and the most negative feedbacks seem to exhibit fairly low familarity with the area, so I recommend an acceptance. The authors are encouraged to incorporate the review comments for further improving the paper.